# Estimated impact of the pneumococcal conjugate vaccine on pneumonia mortality in South Africa, 1999 through 2016: An ecological modelling study

Jackie Kleynhans[1,2]*, Stefano Tempia[2], Kayoko Shioda[3], Anne von Gottberg[1,4], Daniel M. Weinberger[3], Cheryl Cohen[1,2]

**1** Centre for Respiratory Diseases and Meningitis (CRDM), National Institute for Communicable Diseases (NICD) of the National Health Laboratory Service (NHLS), Johannesburg, South Africa, **2** School of Public Health, Faculty of Health Sciences, University of the Witwatersrand, Johannesburg, South Africa, **3** Department of Epidemiology of Microbial Diseases, Yale School of Public Health, New Haven, Connecticut, United States of America, **4** School of Pathology, Faculty of Health Sciences, University of the Witwatersrand, Johannesburg, South Africa

* jackiekleyn@gmail.com

**Data Availability Statement:** The data with aggregated death counts and the R code used for

## Abstract

### Background

Data on the national-level impact of pneumococcal conjugate vaccine (PCV) introduction on mortality are lacking from Africa. PCV was introduced in South Africa in 2009. We estimated the impact of PCV introduction on all-cause pneumonia mortality in South Africa, while controlling for changes in mortality due to other interventions.

### Methods and findings

We used national death registration data in South Africa from 1999 to 2016 to assess the impact of PCV introduction on all-cause pneumonia mortality in all ages, with the exclusion of infants aged <1 month. We created a composite (synthetic) control using Bayesian variable selection of nondiarrheal, nonpneumonia, and nonpneumococcal deaths to estimate the number of expected all-cause pneumonia deaths in the absence of PCV introduction post 2009. We compared all-cause pneumonia deaths from the death registry to the expected deaths in 2012 to 2016. We also estimated the number of prevented deaths during 2009 to 2016. Of the 9,324,638 deaths reported in South Africa from 1999 to 2016, 12·6% were pneumonia-related.

Compared to number of deaths expected, we estimated a 33% (95% credible interval (CrI) 26% to 43%), 23% (95%CrI 17% to 29%), 25% (95%CrI 19% to 32%), and 23% (95% CrI 11% to 32%) reduction in pneumonia mortality in children aged 1 to 11 months, 1 to 4 years, 5 to 7 years, and 8 to 18 years in 2012 to 2016, respectively. In total, an estimated 18,422 (95%CrI 12,388 to 26,978) pneumonia-related deaths were prevented from 2009 to 2016 in children aged <19 years. No declines were estimated observed among adults following PCV introduction. This study was mainly limited by coding errors in original data that

these analyses can be found at https://github.com/weinbergerlab/south_africa_kleynhans.

**Funding:** The author(s) received no specific funding for this work.

**Competing interests:** I have read the journal's policy and the authors of this manuscript have the following competing interests: AvG received research funding from Pfizer and Sanofi unrelated to this project. DMW is PI on a research grant from Pfizer Yale and has received consulting fees from Pfizer, Merck, GSK, and Affinivax. CC has received grant funding from Sanofi Pasteur, US Centers for Disease Control and Prevention and Programme for Applied Technologies in Health awarded to the institute. She has had travel costs reimbursed by Parexel. JK, ST, and KS report no conflicts of interest.

**Abbreviations:** ART, antiretroviral treatment; CrI, credible interval; DHA, Department of Home Affairs; ICD-10, International Classification of Diseases, 10th Revision; IPD, invasive pneumococcal disease; LRTI, lower respiratory tract infections; MCMC, Markov Chain Monte Carlo; PCV, pneumococcal conjugate vaccine; RECORD, REporting of studies Conducted using Observational Routinely collected health Data; Stats SA, Statistics South Africa.

could have led to a lower impact estimate, and unmeasured factors could also have confounded estimates.

## Conclusions

This study found that the introduction of PCV was associated with substantial reduction in all-cause pneumonia deaths in children aged 1 month to <19 years. The model predicted an effect of PCV in age groups who were eligible for vaccination (1 months to 4 years), and an indirect effect in those too old (8 to 18 years) to be vaccinated. These findings support sustaining pneumococcal vaccination to reduce pneumonia-related mortality in children.

## Author summary

### Why was this study done?

- The bacterium, *Streptococcus pneumoniae*, causes a third of childhood pneumonia deaths in sub-Saharan Africa.

- The pneumococcal conjugate vaccine (PCV) has been used in South Africa since 2009 and has been found to be effective in reducing vaccine-serotype invasive pneumococcal disease.

- The effect of PCV on pneumonia mortality has not been assessed in South Africa, or on national scale in Africa.

- We wanted to estimate the impact of this vaccine on all-cause pneumonia mortality in South Africa.

### What did the researches do and find?

- We performed an ecological modelling study using national death registration data in South Africa from 1999 to 2016 to assess the impact of PCV introduction on all-cause pneumonia mortality.

- Due to multiple other changes in South Africa at the time, pneumonia mortality was already decreasing before PCV was introduced. Therefore, we needed to control for these changes while measuring the impact of PCV on pneumonia mortality.

- We did this by creating a synthetic control from multiple other causes of deaths and compared the number of pneumonia deaths after PCV introduction to this synthetic control.

- We estimated that pneumonia mortality reduced between 23% and 33% in children younger than 19 years, with an estimated 18,000 deaths prevented between 2009 and 2016.

### What do these findings mean?

- PCV has contributed in reducing childhood pneumonia mortality in South Africa.

- Vaccination of children with PCV should continue globally and should be expanded to all African countries.

## Introduction

Lower respiratory tract infections (LRTI) were estimated to cause 2.74 million deaths globally in 2015, with 704,000 of those deaths occurring in children <5 years. Forty-eight percent of the global LRTI deaths in children <5 years occur in sub-Saharan Africa, where the slowest rate of decrease in under-5 mortality was seen [1]. In Africa, 33% of childhood pneumonia deaths are attributed to *Streptococcus pneumoniae*, or pneumococcus [2]. The pneumococcal conjugate vaccine (PCV) has potential to greatly reduce childhood mortality. Most assessments of PCV impact on all-cause pneumonia mortality were performed in Latin American and Caribbean countries, with results ranging from no reduction in children <5 years in Brazil, to a 44% reduction in children <1 year in Peru for all-cause pneumonia mortality [3–6]. Data are lacking from Africa, including South Africa.

The 7-valent pneumococcal conjugate vaccine (PCV7) was introduced in the South African Expanded Programme on Immunization in 2009 with no catch-up doses offered to children older than 6 weeks at the time of vaccine introduction (only children aged 6 weeks or younger on 1 April 2009 were eligible for PCV7). The country transitioned to the 13-valent (PCV13) vaccine in 2011 with a limited catch-up where children who had received ≥2 doses of PCV7 received a dose of PCV13 at 18 months and children aged >18 to 36 months received a single dose of PCV13. The vaccine is administered as a 2+1 schedule at 6 weeks, 14 weeks, and a 9-month booster, with 77% coverage for the third dose in 2016 [7]. From 2005 through 2012, there was a 60% decline in the incidence of all-serotype invasive pneumococcal disease (IPD) in South African children aged <2 years [8]. The estimated overall IPD reduction (irrespective of serotype) in children <5 years was 62%, and 77% for pneumococcal meningitis [9].

Estimating the impact of PCV on mortality without the confirmation of the organism is challenging, because of the difficulty in distinguishing the effects of PCV from those of general healthcare improvements [4,5,10]. In South Africa, since 1999, there have been marked changes in mortality as a result of the HIV epidemic and, more recently, improvements in healthcare, especially HIV testing and treatment programmes [8]. South Africa has the fourth highest prevalence of HIV infection globally, with the highest incidence globally, with 1 in 8 new HIV infections occurring in South Africa [11]. Although antiretroviral treatment (ART) coverage was only 18% in 2010, it increased to 65% in 2017 and is expected to reach 73% in 2030 [11]. As a result, HIV-related deaths have already reduced by 10.5% from 2007 to 2017, but the burden remains high with 135,000 HIV-related deaths occurring in 2017 [11].

In this ecological modelling study, we aimed to estimate the impact of PCV introduction on all-cause pneumonia mortality for all ages above 1 month in South Africa, while controlling for changes in mortality resulting from other interventions.

## Methods

### Data sources

We obtained cause of death data from 1999 to 2016 from Statistics South Africa (Stats SA). According to the Births and Deaths Registration Act 1992 (Act No. 51 of 1992), all deaths in South Africa must be registered with the Department of Home Affairs (DHA). Stats SA collects notification forms from DHA for capturing and analysis, which are then summarized in an

annual "Mortality and causes of death in South Africa" report. The reports and data utilized are publicly available from Stats SA [12]. Deaths are coded based on the International Classification of Diseases (ICD), 10th Revision 2016 Edition. Up to 4 causes of death, as well as an "other" and "underlying" cause of death can be captured up to 3-character level (for example, Rheumatic myocarditis (I09.0) would be classified as Other rheumatic heart diseases (I09)). Ten years of prevaccine data (1999 to 2008), a 3-year transition period (2009 to 2011) when PCV7 and PCV13 were introduced and 5 years of postvaccine data (2012 to 2016) were available. We obtained midyear population estimates from Stats SA [13].

## Estimating the impact of PCV on all-cause pneumonia mortality

As laid out in the initial analysis plan (S1 Text), we utilized a synthetic control approach to adjust for disease trends unrelated to the vaccine as described by Bruhn and colleagues [10]. The only change to the analysis plan was to include 2016, as the data became available prior to the start of analysis. In the synthetic control approach, a composite control is constructed by using a weighted average of control conditions. These conditions are weighted based on how well their trend followed the trend of the outcome condition prior to the intervention taking place [14]. Methods are detailed in S2 Text. The data with aggregated death counts and the R code used for these analyses can be found at https://github.com/weinbergerlab/south_africa_kleynhans.

Individual-level mortality data were grouped according to ICD-10 chapter (for exceptions, see S1 Table) by month of death and age group (1 to 11 months, 1 to 4 years, 5 to 7 years, 8 to 18 years, 19 to 39 years, 40 to 64 years, 65 to 79 years, and ≥80 years).

The outcome condition was defined as all-cause pneumonia mortality and consisted of all ICD-10 codes relating to bacterial and viral pneumonia (J12-J18). A death was classified as all-cause pneumonia-related if any of the 6 causes of death fields (cause A-D, other, or underlying cause of death field) included one of these codes. Other causes of death that were found in any of the 6 causes of death fields were classified as controls. If a J12-J18 code existed in any of the 6 causes of death variables, the death would count towards the outcome and not a control. All causes of deaths classified as controls were grouped based on ICD-10 chapter (S1 Table) based on the ICD-10 code listed in any cause of death variable and aggregated to the monthly time scale. A single death could be counted in two or more groups, for example, if C61 was listed as cause A and N39 as cause B, the death counted towards both the C00_D49 and N39 groups. For the controls, we excluded conditions which could have been influenced by the intervention under study (PCV introduction) or individual conditions affected by an intervention which would not also affect pneumonia (e.g., rotavirus vaccine) [10].

The synthetic control model used the Bayesian variable selection (with spike-and-slab priors) to determine the contribution to 38 control diseases (all log-transformed, S1 Table). Each control disease was weighted based on how well it fit the pneumonia deaths time series during 1999 to 2008. Time series that resembled the pneumonia deaths time series were weighted higher, and irrelevant time series were weighted closer to zero. We then fitted a regression model to the monthly prevaccine number of pneumonia deaths (log-transformed) using a Bayesian structural time-series model. This allowed us to quantify uncertainty associated with variable selection and variation in the data. This process generated an equation that described the relationship between the control variables and pneumonia deaths in the absence of the vaccine [10]. A composite control was generated by plugging in the observed values of the control diseases during the post-PCV period into this equation (equation S1). The output from this model provided an estimate for the expected number of pneumonia deaths if the intervention (PCV) had not been introduced for the post-PCV period (the counterfactual). The rate ratio

was calculated by dividing the observed number of deaths by the counterfactual number of deaths during the postvaccine period (2012 to 2016). We drew 10,000 Markov Chain Monte Carlo (MCMC) samples, which allowed us to quantify uncertainty associated with variable selection and variation in the data. The 2.5 and 97.5 percentiles of the MCMC samples provided the 95% credible intervals (CrI). The percentage reduction in mortality during 2012 to 2016 was calculated as 1-rate ratio multiplied by 100. In addition, we estimated the number of all-cause pneumonia deaths prevented by PCV during 2009 to 2016 by subtracting the observed number of deaths from the counterfactual number of deaths.

This study is reported as per the REporting of studies Conducted using Observational Routinely collected health Data (RECORD) Statement (S1 RECORD Checklist).

### Sensitivity analyses

We performed 4 main types of sensitivity analysis (S2 Text) (1) excluding certain control groups due to known anomalies in the source data or to test if the estimated impact of PCV was sensitive to the inclusion of specific control groups; (2) excluding certain years from the analysis to account for known disruptions in the time series data that could potentially influence the estimated impact of PCV (e.g., the 2009 influenza pandemic, abrupt shift in the ICD coding practice); (3) aggregating data by trimester to investigate possible over fitting; and (4) estimating the impact of PCV using a different analytic method (interrupted time series model).

## Results

### Mortality data

During 1999 through 2016, there were 9,324,638 deaths in South Africa reported to Stats SA in all ages (excluding stillbirths), and pneumonia was indicated as one of the causes of death in 12.6% ($n$ = 1,174,504). The proportion of pneumonia-related deaths to all-cause deaths was highest in those aged 1 to 11 months (30%) and lowest in those aged 65 to 79 years (9%, Fig 1). Within the pneumonia deaths, 99% of deaths in those ≥1 year were coded as J18 (pneumonia, organism unspecified), and for children 1 to 11 months, 46% were coded as J18 and 54% as P23 (congenital pneumonia, prior to P23 coding correction, S1 Fig).

The incidence of mortality due to pneumonia increased from 1999 to 2006 and then decreased for all age groups under 65 years (Fig 2). For all ages, there was a 57% reduction in pneumonia mortality from 2006 to 2016 (from 184 to 79 deaths per 100,000 population, Fig 1). When considering the absolute average rate of pneumonia mortality in the post-PCV period (2012 to 2016) to the pre-PCV period (1999 to 2008), the age group with the largest reduction in pneumonia mortality was the 1- to 11-month group, with a 70% reduction (1,091 to 332 deaths per 100,000 population, S2 Table).

### Estimated declines due to introduction of PCVs

Compared to the number of deaths expected in the absence of vaccination, there was a notable reduction in pneumonia mortality for all children <19 years following introduction of PCV. The largest estimated number of deaths prevented (13,168 in 2009 to 2016, 95% CrI 9,292 to 19,479) was seen in children aged 1 to 11 months, with a 33% (95% CrI 26% to 43%) attributable reduction in pneumonia mortality (Table 1, Figs 2 and 3). There was a 23% (95% CrI 17% to 29%) attributable reduction in pneumonia mortality in children aged 1 to 4 years, a 25% (95% CrI 19% to 32%) attributable reduction in children aged 5 to 7 years, and a 23% (95% CrI 11% to 32%) attributable reduction for children aged 8 to 18 years. In total, an estimated

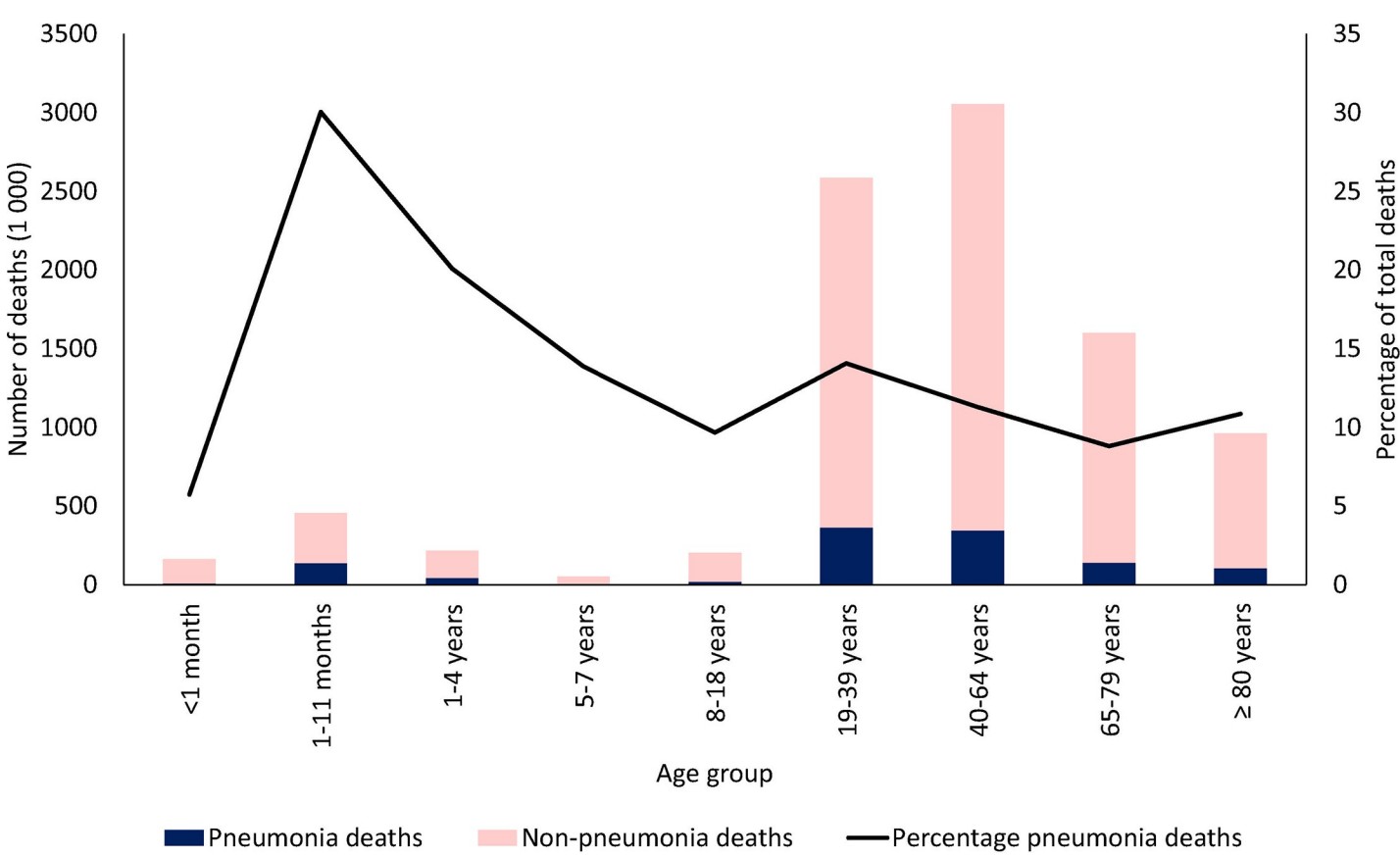

**Fig 1. Total number of pneumonia and nonpneumonia deaths and percentage of deaths codes as pneumonia by age-group in South Africa, 1999–2016.**

18,422 (95% CrI 13,12,388 to 26,978) pneumonia-related deaths were prevented by PCV from 2009 to 2016 in children <19 years. For adults, there were no notable differences between the number of pneumonia-related deaths expected compared to what was observed.

## Sensitivity analyses

Excluding the top 3 controls (S3 Table) in each age group systematically increased effect estimates (S4 Table). When excluding certain chapters or years as input parameters, this did not substantively change the results from the main model, except when we excluded the R chapter (S5 and S6 Tables). Of the 9,149,101 deaths in individuals aged ≥1 month from 1999 to 2016, 16% (*n* = 1,504,016) had an R ICD-10 code in at least 1 of the 6 fields indicating cause of death, with these codes used most commonly in the elderly (S2 and S3 Figs). When this chapter was excluded, a 19% (95% CrI 3% to 32%) reduction of pneumonia mortality was also observed in the 19- to 39-year age group (S5 Table, S4 Fig), with no differences observed in other age categories. When we excluded 1999 to 2005, results were similar for all age groups, except 1 to 11 months where no effect was seen and those 40 to 64 years where a 26% (10% to 36%) reduction in mortality was observed (S5 Table). After excluding the top control from this model, no effect was observed (S7 Table). Excluding 2009 from the analysis, reducing the evaluation period by 2 years or aggregating data by trimester did not influence results (S8 Table). Furthermore, not grouping P codes used in children aged ≥1 month, and not grouping D50-D89 with A10-B99 made no difference in estimates (S8 Table).

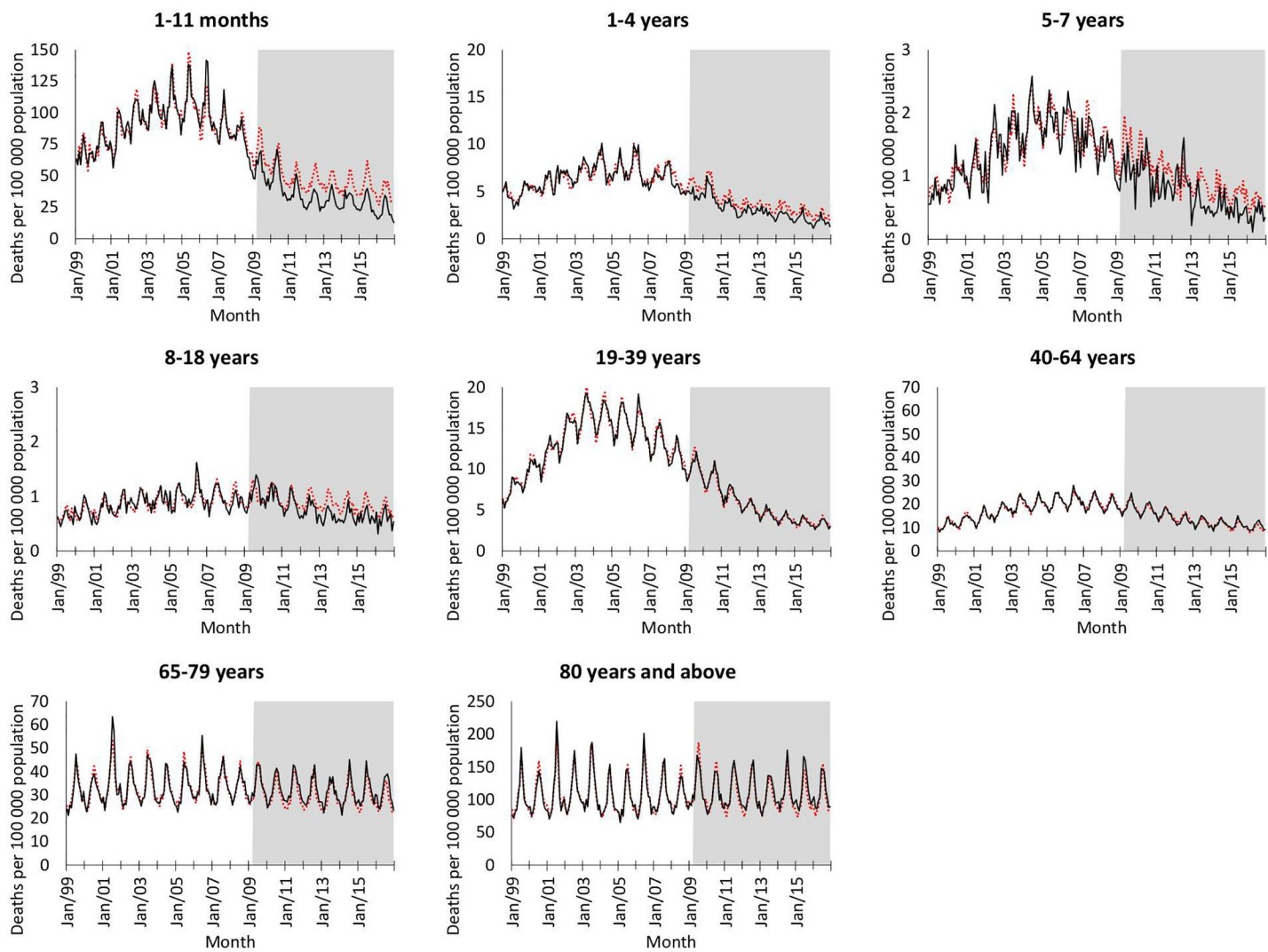

**Fig 2. Time series of all-cause pneumonia mortality per 100,000 population and counterfactual estimates using synthetic controls model by age group, 1999–2016, South Africa.** Black, solid lines denote actual all-cause pneumonia. Red, dashed lines denotes counterfactual estimates. Intervention period indicated in grey. Scale different across age groups.

**Table 1. Number of estimated deaths prevented in children <19 years since PCV introduction, South Africa, 2009–2016.**

|  | 1–11 months (95% CrI) | 1–4 years | 5–7 years | 8–18 years |
|---|---|---|---|---|
| 2009 | **1,539 (1,034 to 2,119)** | **425 (265 to 592)** | **77 (32 to 131)** | −67 (−175 to 49) |
| 2010 | **1,309 (841 to 1,866)** | **405 (278 to 518)** | **67 (43 to 95)** | 43 (−86 to 149) |
| 2011 | **1,412 (974 to 2,068)** | **408 (331 to 507)** | **106 (84 to 134)** | 117 (−55 to 191) |
| 2012 | **2,072 (1,602 to 2,839)** | **373 (271 to 501)** | −34 (−49 to 6) | **223 (52 to 321)** |
| 2013 | **1,740 (1,214 to 2,668)** | **363 (278 to 495)** | **126 (108 to 148)** | **342 (212 to 462)** |
| 2014 | **1,467 (979 to 2,524)** | **504 (408 to 612)** | **99 (81 to 122)** | **276 (134 to 414)** |
| 2015 | **1,929 (1,407 to 2,769)** | **410 (320 to 538)** | **107 (90 to 127)** | **231 (91 to 395)** |
| 2016 | **1,700 (1,241 to 2,625)** | **352 (243 to 520)** | **87 (71 to 104)** | **215 (69 to 367)** |
| Cumulative | **13,168 (9,292 to 19,479)** | **3,239 (2,395 to 4,283)** | **635 (459 to 868)** | **1,380 (242 to 2,348)** |

CrI, credible interval. Significant predictions in bold.

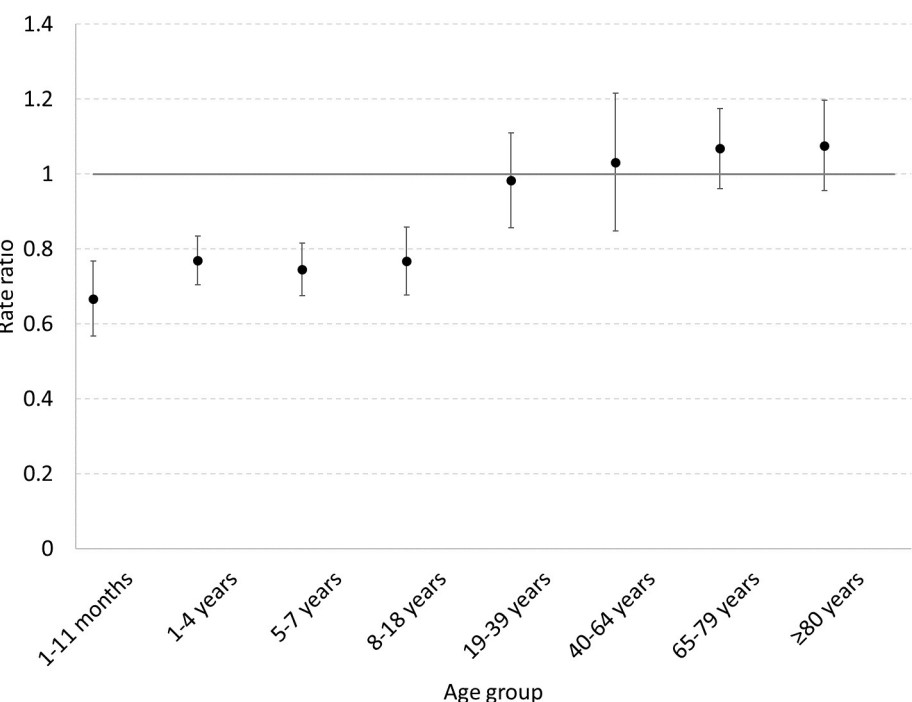

**Fig 3. Changes in deaths for all-cause pneumonia mortality (rate ratio) using the synthetic control model by age group in the postvaccine period (2012–2016), South Africa.** Circle indicates rate ratio estimate and protruding lines the 95% credible interval.

For all age groups with reductions in pneumonia mortality, the interrupted time series analysis predicted a greater reduction in deaths than the synthetic control model (S9 Table).

## Discussion

We estimated a substantial reduction in all-cause pneumonia mortality in children aged 1 month to <19 years in South Africa after the introduction of PCV in 2009. After adjusting for other factors which have impacted pneumonia mortality, we estimated that PCV introduction was associated with mortality reductions in age groups eligible for vaccination, with a reduction of 33% and 23% of pneumonia mortality in children aged 1 to 11 months and 1 to 4 years, respectively. We also estimated a 23% reduction in mortality in children aged 8 to 18 years who were too old to be eligible for vaccination. The estimated effect of the vaccine in children eligible for vaccination was observed from 2009 onwards, with the indirect effect of the vaccine in those not eligible for vaccination evident from 3 years after the vaccine introduction in 2009. These reductions are in line with the predicted 25% decline in childhood pneumonia mortality after PCV introduction [15].

Africa is the WHO region with the highest burden of under-5 mortality, and where most pneumonia deaths occur [16]. Our data suggest that the introduction of PCV prevented a substantial number of deaths in the 8 years after its introduction in South Africa, with approximately 18,400 deaths prevented from 2009 to 2016 in children aged 1 month to <19 years. Reductions in the number of deaths were observed from the first year of PCV7 introduction in 2009 in children aged 7 years and younger, and only from 2012 in children aged 8 to 18 years. This corresponds to the reduction of invasive pneumococcal disease incidence already observed from 2009 in South Africa, especially in children younger than 2 years [8].

This estimated reduction in all-cause pneumonia mortality was observed against a backdrop where coverage of a third dose of PCV in South Africa was initially low with 62% in 2011, increased to 85% in 2014 and 2015, and decreased again to 77% in 2016 [7]. Higher vaccine coverage in South Africa may lead to a larger reduction in pneumonia deaths, as higher levels of uptake of the vaccine was associated with larger declines in all-cause pneumonia hospitalizations [17] and pneumonia mortality [18] in Brazil.

The indirect effect of the vaccine has been shown to reduce IPD, all-cause pneumonia, and pneumonia mortality in adults in some countries, whereas no indirect effect has been observed in others [19]. We did not observe any reductions in mortality among adults despite previously demonstrated reductions in adult IPD [8]. Pneumococcus may contribute to a lower proportion of pneumonia in adults in South Africa compared to children, and therefore, the effect might be too small to identify using an ecological study, or replacement of nonvaccine serotype disease may be occurring as reported in some countries [19]. Vaccine replacement has been described for IPD in South Africa [20]. If pneumonia-related deaths were coded under R codes, we could have biased our estimate of PCV impact towards the null by including the R chapter as a control. On sensitivity analysis by excluding the R chapter, a 19% reduction of pneumonia mortality was estimated in the 19- to 39-year age group, where the highest burden of HIV infection and HIV-related mortality lies in South Africa [21]. The model scenario where we exclude the R chapter from the composite control used in the sensitivity analysis might not control adequately for improvements in HIV programmes in this age group if some of the HIV-related deaths were coded under the R codes.

There has been an overall trend of pneumonia mortality reductions globally, with a 94% reduction in pneumonia mortality from 56 countries in children <5 years from 1960 to 2012 [22]. In countries where mortality rates were low prior to PCV introduction, the impact of PCV on mortality has been negligible [23]. Overall, these large declines were not only attributable to the introduction of the PCV, but likely due to economic improvement in countries, leading to better socioeconomic conditions, better interventions to prevent childhood mortality (for example, the integrated management of childhood illness) [24], the introduction of measles vaccine, and the implementation of the fourth millennium development goal, i.e., "Reduce child mortality by two-thirds relative to 1990" [22,25]. Specifically, in South Africa, the reduction in mortality started in 2006, largely related to increased coverage of HIV detection, treatment, and prevention programmes [26]. It was estimated that ART prevented 1.72 million HIV-related deaths in South African adults between 2000 and 2014 alone [27]. When we reduced the prevaccine period for the analysis, using data from 2006 onwards to exclude the period in which pneumonia mortality was increasing, most likely attributable to South Africa's HIV epidemic [27]; we observed no effect of PCV in those aged 1 to 11 months and 26% reduction in the 40 to 64 year olds, results unique to this specific sensitivity analysis with no similar signals in any of the other sensitivity analyses. It is interesting to note that the sudden change in R coding practices for the 1 to 11 months old occurred in 2006, but we do not believe this to be the reason for this difference as excluding the R chapter from this same analysis still did not result in a measured effect. In the backdrop of upscaling prevention of mother-to-child HIV transmission, and by not considering prior years, the added effect of PCV may have been too small to measure using an ecological study. The effect observed in the 40- to 46-year-old group when excluding pre-2006 data could possibly be due to an artifact of the control choice.

Although PCV is one of the most expensive vaccines included in routine childhood vaccination programmes, it has been shown to be a cost-effective option to reduce childhood morbidity and mortality, especially in Africa [28]. It is estimated that if PCV were used in every country in the world, it would come at a cost of $15.5 million USD but would save around

400,000 lives and prevent 54.6 million disease episodes annually [28]. This makes PCV a very powerful tool to combat childhood mortality. There are still 9 African countries in which PCVs are yet to be introduced [29], and others that will no longer be eligible for GAVI support and need to fund vaccination from national budgets [28]. Studies on the impact of the vaccine are therefore crucial to understand the possible benefits of introducing or maintaining pneumococcal vaccination in these countries.

To our knowledge, this is the first study on the impact of PCV on pneumonia mortality in all age groups on a national scale in Africa. A 16% reduction in all-cause childhood mortality was observed during a randomized control trial evaluating PCV9 in Upper and Central River Division of The Gambia [30]. Using a similar methodological approach in Brazil, Schuck-Paim and colleagues found that on a national level, there were no pneumonia mortality reductions attributable to PCV in children aged <5 years (on a background of substantial reductions due to economic improvements). PCV likely contributed to a reduction in pneumonia mortality of between 16% and 24% in Brazilian municipalities with a high percentage of extreme childhood poverty and mothers without primary education [4]. The South African model used different controls in the synthetic model as applied in Brazil. The top 3 controls included in the synthetic control model in Brazil in the 3 months—5 years old were all non-PCV-related deaths, Intestinal infectious diseases (A00-A09), and Ill-defined and unknown causes of mortality (R95-R99). In our analysis, the R chapter was the third top control for the 1- to 11-month group, but no other similarities were observed in the 1- to 4-year group. This highlights how setting-specific trends may influence the utility of certain control groups when using the synthetic control approach. Our results are comparable to what was found in some Latin American and Caribbean countries, where the reduction in pneumonia mortality ranged from 11% in Mexico to 35% in Peru in children aged 2 months to 5 years [5].

In Soweto, South Africa, pneumonia-related hospitalizations reduced by half from 2006 to 2014 in HIV–uninfected children <5 years following the introduction of PCV in 2009 [31]. A modelling study in South Africa estimated a 60% reduction in nonbacteremic pneumococcal pneumonia after the introduction of PCV, assuming pneumococcus made up the etiological agent for 33% of pneumonia and influenza-related deaths in children <5 years [32], which is consistent with the reduction range of 23% to 39% that we estimated in this study. The PERCH study, mostly conducted in countries with established PCV programmes, however found only 5% of pneumonia cases to be attributable to pneumococcus [33].

The synthetic control approach is more suitable than interrupted time series analysis for assessment of PCV impact on hospitalisations [10] and mortality [4,5]. Interrupted time series does not account for changes in trends in the evaluation period not due to the intervention which may reduce the validity of the estimation of impact [34] which was suggested by the overestimation of impact when the interrupted time series method was applied to these data. In contrast, the synthetic control approach incorporates several time series, which are weighted based on their fit to the time series under study before the intervention, allowing for adjustment for unmeasured bias and confounding. Specifically, in South Africa, where the upscaling of ART utilization had a large effect on mortality trends, the synthetic control approach enabled us to estimate the impact of PCV accounting for nonlinear background mortality trends.

This study had limitations. We corrected coding to the best of our ability; however, some residual errors might have remained. On sensitivity analysis, no notable differences were observed in the results when data were used without coding corrections. Residual coding errors would likely have led to pneumonia deaths coded to other chapters and an underestimation of the vaccine impact due to vaccine effects being observed in both the outcome and control and therefore biasing the reduction to zero. Our selection of controls was limited to the

data available, but results were robust to changes in controls on sensitivity analysis, suggesting that this approach was able to adequately control for sociodemographic and health system-related changes. The vaccine impact evaluated here included the reduction of PCV7 and PCV13 serotype-related deaths, potential increase in nonvaccine serotypes-related deaths (replacement), the direct effect on vaccinated children, and the indirect effect on the nonvaccinated population, and we were not able to distinguish these effects individually using this ecological approach. Shortening of the analysis period prior to the increase of nonvaccine serotypes in South Africa was not possible as the increase in nonvaccine serotypes was already signaled in invasive pneumococcal disease in 2010 [20]. We only used monthly aggregated data and did not assess the model using weekly or yearly aggregated data. Different aggregation could possibly affect estimates. We were also not able to differentiate between direct and indirect vaccine effects in the 5- to 7-year-old age group as this grouped contained a mix of individuals who by 2016 might not have been vaccinated or could have been vaccinated as part of a catch-up campaign or routine vaccination schedule.

## Conclusion

This study estimated that the introduction of PCV in South Africa resulted in a notable reduction of all-cause pneumonia mortality in children eligible for vaccination, as well as those not eligible for vaccination, which is in-line with previous projections. PCV was estimated to have prevented approximately 2,300 deaths annually from 2009 to 2016 in children <19 years. To our knowledge, these are the first national-level data from an African country showing the impact of PCV on pneumonia mortality. These data support sustaining pneumococcal vaccination and the expansion of vaccination to all African countries.

## Supporting information

**S1 RECORD Checklist. The RECORD statement.**
(PDF)

**S1 Text. Analysis plan.**
(PDF)

**S2 Text. Detailed methods.**
(PDF)

**S1 Fig. Pneumonia-related deaths by age and year in South Africa, 1999–2016.**
(PDF)

**S2 Fig. Percentage of deaths with a cause of death from ICD-10 R chapter indicated as cause of death (main, underlying, or other cause of death).** Data uncorrected for P chapter miscoding prior to 2006.
(PDF)

**S3 Fig. Percentage and number of deaths with a cause of death from ICD-10 R chapter indicated as cause of death (main, underlying, or other cause of death) for children aged 1–11 months.** Data uncorrected for P chapter miscoding prior to 2006.
(PDF)

**S4 Fig. Time series of all-cause pneumonia mortality per 100,000 and counterfactual estimates using synthetic controls model in those aged 19–39 years, 1999–2016, South Africa.** Main result (left) and the sensitivity analysis where R chapter was removed (right). Black, solid lines denote actual all-cause pneumonia. Red, dashed lines denotes counterfactual estimates.

Intervention period indicated in grey.
(PDF)

**S1 Table. Description of ICD codes used for outcome and composite control.**
(PDF)

**S2 Table. Rate of observed and predicted (counterfactual) deaths per 100,000 population due to all-cause pneumonia in the prevaccine period (1999–2008) and the postvaccine period (2012–2016). Significant predictions in bold.** *Observed rate in 2012–2016 compared to observed rate in 1999–2008. †Observed rate in 2012–2016 compared to predicted rate in 2012–2016, significant differences in bold.
(PDF)

**S3 Table. Top 3 control groups included in main synthetic control analysis by age.**
(PDF)

**S4 Table. Sensitivity analysis of changes in deaths for all-cause pneumonia mortality (rate ratio) by removing the controls with highest inclusion probabilities, in the postvaccine period (2012–2016), South Africa.** Rate ratio (RR), 95% credible interval (CrI) in brackets, significant predictions in bold.
(PDF)

**S5 Table. Sensitivity analysis of changes in deaths for all-cause pneumonia mortality (rate ratio) by removing certain controls, in the postvaccine period (2012–2016), South Africa.** Rate ratio (RR), 95% credible interval (CrI) in brackets, significant predictions in bold.
(PDF)

**S6 Table. Sensitivity analysis of changes in deaths for all-cause pneumonia mortality (rate ratio) on unadjusted data (no reclassification of P (neonatal) codes in children aged 1–11 months, and not grouping D50-D89 with A10-B99), in the postvaccine period (2012–2016), South Africa.** Rate ratio (RR), 95% credible interval (CrI) in brackets, significant predictions in bold.
(PDF)

**S7 Table. Sensitivity analysis of changes in deaths for all-cause pneumonia mortality (rate ratio) on 2006–2016 analysis, by removing the controls with highest inclusion probabilities, in the postvaccine period (2012–2016), South Africa.** Rate ratio (RR), 95% credible interval (CrI) in brackets, significant predictions in bold.
(PDF)

**S8 Table. Sensitivity analysis of changes in deaths for all-cause pneumonia mortality (rate ratio) by removing certain years, or aggregating by trimester, in the postvaccine period (2012–2016), South Africa.** Rate ratio (RR), 95% credible interval (CrI) in brackets, significant predictions in bold.
(PDF)

**S9 Table. Changes in deaths for all-cause pneumonia mortality (rate ratio) from synthetic control (SC) and interrupted time series (ITS) analysis, in the postvaccine period (2012–2016), South Africa.** Rate ratio (RR), 95% credible interval (CrI) in brackets, significant predictions in bold.
(PDF)

## Acknowledgments

We would like to thank Statistics South Africa for the use of the mortality and denominator data.

## Author Contributions

**Conceptualization:** Stefano Tempia, Anne von Gottberg, Cheryl Cohen.

**Data curation:** Jackie Kleynhans, Stefano Tempia, Cheryl Cohen.

**Formal analysis:** Jackie Kleynhans.

**Methodology:** Kayoko Shioda, Daniel M. Weinberger.

**Software:** Kayoko Shioda, Daniel M. Weinberger.

**Writing – original draft:** Jackie Kleynhans.

**Writing – review & editing:** Jackie Kleynhans, Stefano Tempia, Kayoko Shioda, Anne von Gottberg, Daniel M. Weinberger, Cheryl Cohen.

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
