## [Editor Report · Decision Letter 0]

1 Sep 2020

Dear Dr Kleynhans, 

Thank you for submitting your manuscript entitled "Impact of the pneumococcal conjugate vaccine on pneumonia mortality in South Africa, 1999 through 2016" for consideration by PLOS Medicine.

Your manuscript has now been evaluated by the PLOS Medicine editorial staff as well as by an academic editor with relevant expertise and I am writing to let you know that we would like to send your submission out for external peer review.

Kind regards,

Thomas J McBride, PhD

Senior Editor

PLOS Medicine

---

## [Decision Letter · Decision Letter 1]

30 Sep 2020

Dear Dr. Kleynhans,

Thank you very much for submitting your manuscript "Impact of the pneumococcal conjugate vaccine on pneumonia mortality in South Africa, 1999 through 2016" (PMEDICINE-D-20-04020R1) for consideration at PLOS Medicine. 

Your paper was evaluated by a senior editor and discussed among all the editors here. It was also discussed with an academic editor with relevant expertise, and sent to three independent reviewers, including a statistical reviewer (#1). The reviews are appended at the bottom of this email and any accompanying reviewer attachments can be seen via the link below:

[LINK]

In light of these reviews, I am afraid that we will not be able to accept the manuscript for publication in the journal in its current form, but we would like to consider a revised version that addresses the reviewers' and editors' comments. Obviously we cannot make any decision about publication until we have seen the revised manuscript and your response, and we plan to seek re-review by one or more of the reviewers. 

We expect to receive your revised manuscript by Oct 21 2020 11:59PM. Please email us (plosmedicine@plos.org) if you have any questions or concerns.

We look forward to receiving your revised manuscript. 

Sincerely,

Emma Veitch, PhD

PLOS Medicine

On behalf of Tom McBride, PhD, Senior Editor, 

PLOS Medicine

plosmedicine.org

Academic editor comments -

"I would recommend major revision, with more detail on methods and the exact way they constructed the synthetic cohort would help. Reviewer 2 has asked for minor revisions but her comments are tougher than the others. I also think they need to explain the potential impact of the PCV 13 vs 10 transition as well as the potential increase of non-vaccine serotypes in SA".

Internal editorial requests (formatting/writeup) 

*We'd recommend revising the title per PLOS Medicine's usual style, normally this includes some indication of the study design/methodological approach in the subtitle of the title (usually after a colon). In this case would "ecological analysis" or "modelling study" be appropriate (the authors clearly are best placed to decide here on what is most indicative).

*At this stage, we ask that you include a short, non-technical Author Summary of your research to make findings accessible to a wide audience that includes both scientists and non-scientists. The Author Summary should immediately follow the Abstract in your revised manuscript. This text is subject to editorial change and should be distinct from the scientific abstract. Please see our author guidelines for more information: https://journals.plos.org/plosmedicine/s/revising-your-manuscript#loc-author-summary

*If possible please reformat the citation style into PLOS Medicine's format (should be straight forward if using referencing software) - this should use callouts formatted as sequential numerals in square brackets (not superscript). Many thanks

*Minor typo in the discussion (trail not trial) - 

"To our knowledge, this is the first study on the impact of PCV on pneumonia mortality in all age groups on a national scale in Africa. A 16% reduction in all-cause childhood mortality was observed during a randomized control trail evaluating PCV9 in Upper and Central River Division of The

Gambia.32"

*We'd ask the authors to clarify in the revised paper whether the analytical approach followed in this study corresponded to one laid out in a prospective protocol or analysis plan. Please state this (either way) early in the Methods section.

Comments from the reviewers:

Reviewer #1: Statistical review

This paper reports an analysis investigating to what extent the introduction of pneumococcal conjugate vaccine reduced the mortality due to pneumonia in South Africa. An interesting analysis approach is used, based on using a synthetic control group from other mortality reasons. My most major comment (see comment 3) is that I think more information is required on the method.

I have some comments, listed below:

1. Abstract: it would be useful to add a brief description of the analysis methods (i.e that it is Bayesian).

2. Page 4 - it may be common language in vaccine settings but I didn't follow what 'no catch up' and 'limited catch up' meant. Is this referring to not giving extra doses to children who were slightly too old at the introduction of the vaccination programme?

3. Statistical methods: I think the authors give intuition over the approach they have used, and provide full code for reproducing the results, which is commendable. I think some of the steps of the process could do with fleshing out a bit in the paper (or supplementary material). I think the text in the second paragraph of page 7 (starting 'We fitted a regression…') requires some more details:

a) 'time series of all possible controls as covariates' - I did not follow how many covariates this would be or how they would be produced. Is this part of the model building, with each possible cause of death a potential control?

b) I think more details over the Bayesian variable selection model should be provided - is this some sort of Bayesian sparse regression model?

c) Does the BVS model result in uncertainty? It's implied that a deterministic equation is used for the expected number of deaths. Or is this done deterministically within each MCMC sample with the uncertainty dealt with that way?

d) Is the relationship between the selected control conditions and pneumonia stable over time in the pre-vaccination period? Would this have an impact on the results and level of uncertainty in the post-vaccination period?

4. Results: The authors present plenty of sensitivity analyses. One set of analyses excluded a couple of different years. I didn't really follow the rationale for picking those years. Given that pneumonia mortality increased until 2006 and then decreased, I would have been interested to see whether excluding all years before 2006 led to any changes to the results.

James Wason

Reviewer #2: In this paper the authors analyze the effectiveness of PCV 10 on pneumonia and mortality among all age groups in South Africa. The paper is clearly written and the data presented are sound. I have only a few remarks:

I could not follow the link to "Oliveira LHd, Shioda K, Valenzuela MT, et al. Declines in Pneumonia Mortality Following the Introduction of Pneumococcal Conjugate Vaccines in Latin American and Caribbean Countries. (05/31/2019 20:53:28) Available at SSRN: https://ssrncom/abstract=3398536 2019." Please update and check all other links to see if they are still valid

Apart from rotavirus vaccines, were there any other vaccines introduced in the country during the study period, considering all age groups? What could be there potential effect?

When explaining the selection of the age-strata used in the study, we are given the impression that 5-7-year-old kids could only have been vaccinated in catch up campaigns. However, as the years accumulate after vaccination start, most of the vaccinated ones in this age group must have received their vaccination as scheduled by the NIP in previous years of their life. Please comment the vaccine cohort effect and its implication for the interpretation of results.

"Control causes of deaths were grouped based on ICD-10 chapter". In order to do this grouping, did you use the "underlying cause of death"? Clarify.

The following paragraph is not clear and has no reference: "We weighted the controls using Bayesian variable selection based on their ability to jointly fit the trend of the outcome variable (pneumonia deaths) in the pre-vaccine period. We drew 10 000 samples using Markov Chain Monte Carlo (MCMC). This process generates an equation that describes the relationship between the control variables and pneumonia deaths in the absence of the vaccine." 

If the reference to this sentence is that of Bruhn et al mentioned several paragraphs before, it should be referenced again. The methodology, even as described in this paper, is very complex and should be more easily explained to the broad readership of this journal. How were the controls weighted? Did you also control for prior trends and seasonality? 

In fact, many vaccine impact studies have been done without the synthetic control approach. Are all these studies biased and wrong? Apart from the sensitivity analysis which you have already performed, would it be of interest to see the impact measurements obtained with and without the use of this methodology to see how different they are and to be able to recommend more firmly this methodology for future studies? Please comment and add to the discussions. 

On the same topic, you mention in your discussion that "Our selection of controls was limited to the data available, but results were robust to changes in controls on sensitivity analysis, suggesting that this approach was able to adequately control for sociodemographic and health system related changes." Could you explain this further? My question is: could the fact that your results did not change that much when you performed extensive sensitivity analyses, excluding big groups of causes from your control group, indicate the opposite, i.e. that these analyses were not being able to adequately control for sociodemographic and health system related changes?

I did not understand what was done for the P code: "For the purpose of this study, all P codes used in children aged >1 month were grouped with non-P codes where possible, as previously described". When you say there was "a systematic misclassification of ICD codes among children aged <1 year into the P code groups", could have been pneumonia J12-J18 deaths among those coded with P codes? Please explain in a clearer format.

The percentage of deaths with a cause of death from ICD-10 R chapter indicated as cause of death (main, underlying or other cause of death) increased abruptly for children aged 1-11 months in 2006. What proportion of those were classified as cases or controls, pre and post vaccination? This needs to be further explained. What is the influence of this change in the results for the target age group? I do not think the exclusions of the R codes from the controls tackles this issue. 

It was not clear to me why you performed sensitivity analyses excluding the following years: 2001 (two-years from start of data), 2008 (one year prior to introduction) and 2015 and 2016 (reducing evaluation period by 2 years). What does the fact that these exclusions did not change the results tell us? Please comment and add to the text. 

You are analyzing this time-series several years after vaccine introduction. You suggest some possibilities for the lack of indirect effect found in older individuals. But what about serotype replacement? Could it be reducing the observed vaccine effects in younger individuals and contributing to the lack of effect in adults? A word on that would be welcomed.

I think we can assume that the impact of vaccination could be greater with a greater vaccination coverage. Perhaps a word on this could be added to the discussion/conclusion.

Reviewer #3: The study "Impact of the pneumococcal conjugate vaccine on pneumonia mortality in South Africa, 1999 through 2016" is aiming to fill a glaring gap in our understanding of PCV impact in Africa; that is the impact of PCV on (pneumonia) mortality. The study is well written and generally conducted well. The ecological analysis however, has to fully rely on a non-mechanistical machine learning like approach because of major changes in pneumonia mortality observed at baseline that likely have continued during the period of vaccination. I think the synthetic control methodology is the best approach to estimate impact in this situation but my main concern is that we will need to completely rely on it and hence I would like to see a number of additional sensitivity analyses to ensure the robustness of the approach.

- The synthetic control is a powerful analysis and one of the very few ways to extract a signal off the noisy data at hand. But it comes with a number of caveats. These include: 1) covariate selection is not hypothesis driven 2) which in combination with the very detailed data at hand makes it prone to over fitting and 3) it is not necessarily clear that the relationship with pneumonia mortality would stay the same in the post vaccination era. The authors do a number of helpful sensitivity analysis but I think it would be good to also do the following:

- A Morris Method like approach where you re-run the analysis a couple of hundred time and for each run you randomly select only 2/3 of the available set of covariates

- To evaluate the balance of making best use of potential signals and limiting the potential for over-fitting I would like to see the analyses re-done but on data in weekly as well as annual aggregation

- Further, some of the authors have used the method in another setting, Brazil, before. While relevant covariates may differ to SA, some may actually have a more engrained role. I suggest to a) report not only the ICD codes used in the control but reported on their contribution to the control, b) compare those to the ICD codes that contributed to the Brazil analysis and c) rerun the analysis but only include the covariates that were deemed relevant for the Brazil analysis to see whether the control can still be fit.

- On page 6 the grouping of age groups into vaccine eligible needs a bit more nuancing. E.g the 1m-4y old age group has only been completely age eligible for vaccination from 2013. Similarly, Table 1 is quite puzzling in that for the age groups 1-4y and 5-7y there is basically the same annual number of deaths prevented from the start of PCV vaccination. One would rather have expected that this would gradually build up as such cohorts get vaccinated and herd immunity kicks in. Could the authors compare this with the timelines of effects on IPD to get a better understanding whether this makes sense or not?

- The authors assess the impact of PCV. However, PCV7 was replaced with PCV13 in 2011 and in principle their analysis should allow to estimate the additional effect of PCV13 on top of PCV13 which would be interesting to try and tease out. It looks like there may be a signal in the <12m olds but it may be more difficult in the older children. As effects would overlap with the gradual build up of herd immunity.

[LINK]

---

## [Decision Letter · Decision Letter 2]

23 Dec 2020

Dear Dr. Kleynhans,

Thank you very much for re-submitting your manuscript "Impact of the pneumococcal conjugate vaccine on pneumonia mortality in South Africa, 1999 through 2016: an ecological modelling study" (PMEDICINE-D-20-04020R2) for review by PLOS Medicine.

I have discussed the paper with my colleagues and the academic editor and it was also seen again by all three reviewers. I am pleased to say that provided the remaining editorial and production issues are dealt with we are planning to accept the paper for publication in the journal.

[LINK]

We expect to receive your revised manuscript within 1 week. I realize I am sending this during the holiday season, so this timeline is flexible. Please email us (plosmedicine@plos.org) if you have any questions or concerns.

We look forward to receiving the revised manuscript by Dec 30 2020 11:59PM.   

Sincerely,

Thomas McBride, PhD

Senior Editor 

PLOS Medicine

plosmedicine.org

Requests from Editors:

1- Thank you for including your analysis plan, please make this a standalone supplementary file.

2- Please ensure that the study is reported according to the RECORD guideline, and include the completed RECORD checklist as Supporting Information. Please add the following statement, or similar, to the Methods: "This study is reported as per the REporting of studies Conducted using Observational Routinely-collected health Data (RECORD) Statement (S1 Checklist)."

The RECORD guideline can be found here: https://www.equator-network.org/reporting-guidelines/record/

3- As this is a modelling study based on observational data, please edit causal language throughout the manuscript, beginning with the title, which can be edited to “Estimated impact…”, as well as editing words such as “effect” at lines 51-2, 242-3, 341-2, and “prevented" at line 352. Please do check for any language we may have missed.

4- At line 47-78, please remove “(although corrected to the best of our ability)”, and add a point to note the possible influence of unmeasured confounders.

5- Please add “This study found that…” or similar language in the Abstract conclusions and the Discussion conclusion.

6- Line 58, “pneumonia”

7- The second section of the Author summary could include a description of the study design (using national death registration data in South Africa from 1999-2016 to assess the impact of PCV introduction on all-cause pneumonia mortality) before describing the synthetic control.

8- Please move the reference callouts to before the punctuation. 

9- In the Discussion Conclusion, please temper the claim of primacy, e.g., “To our knowledge, these are the first national-level data…”

10- The Contributors, Declaration of interest, Funding statements can go from the end of the text and into the submission metadata.

Comments from Reviewers:

Reviewer #1: Thank you to the authors for addressing my previous comments well. I have no further issues to raise.

Reviewer #2: The manuscript has improved substantially after the reviewer's comments. The authors have replied satisfactorily to all the issues that had been raised. 

Reviewer #3: The authors have addressed some of my comments but have decided to not run any of the suggested sensitivity analyses. As a result, I remain a bit concerned about the robustness of the inference. In the end you find correlations in any large enough data set which doesn't necessarily mean that those correlations are predictive of future changes. In more detail:

- While I think the Morris method is more robust I can see that the author's approach to exclude the top three controls is not too dissimilar, so that's okay I guess

- The authors argue their case to not use different data aggregation. If the inference indeed is only valid if run on months that still does tell the reader something about the robustness of the inference and should hence be included as a discussion point I think. 

- The important covariates to model the outcome were very different in this analysis compared to a similar one done for Brazil. While it is entirely possible that this is a true difference it also should be included as a discussion point. Similarity of important covariates would have been somewhat reassuring that there is some structure in what predictors for pneumococcal disease changes are. Similarly it would have been reassuring to see that a similar model to the Brazilian one would have led to similar results; though I agree that absence of such doesn't necessarily invalidate the results in this study

[LINK]

---

## [Editor Report · Decision Letter 3]

12 Jan 2021

Dear Dr Kleynhans, 

On behalf of my colleagues and the Academic Editor, Zulfiqar A. Bhutta, I am pleased to inform you that we have agreed to publish your manuscript "Estimated impact of the pneumococcal conjugate vaccine on pneumonia mortality in South Africa, 1999 through 2016: an ecological modelling study" (PMEDICINE-D-20-04020R3) in PLOS Medicine.

PRESS

Sincerely, 

Thomas J McBride, PhD 

Senior Editor 

PLOS Medicine